# IFN-γ Induces PD-L1 Expression in Primed Human Basophils

**DOI:** 10.3390/cells11050801

**Published:** 2022-02-25

**Authors:** Srinivasa Reddy Bonam, Camille Chauvin, Mano J. Mathew, Jagadeesh Bayry

**Affiliations:** 1Institut National de la Santé et de la Recherche Médicale, Centre de Recherche des Cordeliers, Sorbonne Université, Université de Paris, 75006 Paris, France; bsrpharmacy90@gmail.com (S.R.B.); camille.chauvin@inserm.fr (C.C.); 2EFREI, 94800 Villejuif, France; mano.mathew@efrei.fr; 3Department of Biological Sciences & Engineering, Indian Institute of Technology Palakkad, Palakkad 678623, India

**Keywords:** human basophils, PD-L1, IFN-γ, IL-3, IL-4, IL-13, *IFNGR2*

## Abstract

Programmed death-ligand 1 (PD-L1) plays a key role in maintaining immune tolerance and also in immune evasion of cancers and pathogens. Though the identity of stimuli that induce PD-L1 in various human innate cells and their function are relatively well studied, data on the basophils remain scarce. In this study, we have identified one of the factors, such as IFN-γ, that induces PD-L1 expression in human basophils. Interestingly, we found that basophil priming by IL-3 is indispensable for IFN-γ-induced PD-L1 expression in human basophils. However, priming by other cytokines including granulocyte-macrophage colony-stimulating factor (GM-CSF) and thymic stromal lymphopoietin (TSLP) was dispensable. Analyses of a published microarray data set on IL-3-treated basophils indicated that IL-3 enhances *IFNGR2*, one of the chains of the IFNGR heterodimer complex, and *CD274*, thus providing a mechanistic insight into the role of IL-3 priming in IFN-γ-induced PD-L1 expression in human basophils.

## 1. Introduction

Despite being a minor population of myeloid cells, basophils play a vital role in the immune system [1,2,3,4]. The pertinent role played by the basophils in the protection against helminth infections, and in the pathogenesis of asthma and other diseases is well documented [5]. More than ever, much interest is focused on understanding the role of various granulocytes in the immunopathology of various diseases, particularly diseases of the respiratory system, e.g., coronavirus disease (COVID-19) [6]. Several recent reports on COVID-19 have deciphered changes in the frequency of basophils, and surface molecules associated with their activation and degranulation [6,7]. On the other hand, very few reports have delineated the expression of programmed death-ligand 1 (PD-L1) in basophils [6]. However, the nature and identity of the stimuli that induce PD-L1 in basophils are not yet known. Current data show that PD-L1 expression could be induced in innate cells by several factors, e.g., pro-inflammatory cytokines or pathogen-derived stimuli [8,9,10,11]. Of note, IFN-γ induces PD-L1 expression in human macrophages, neutrophils [12], and several cancer cells [13,14]. Therefore, in this report, we determined the effect of IFN-γ on PD-L1 expression in human basophils. Consequently, we found that IFN-γ induces PD-L1 expression in basophils. Interestingly, IFN-γ alone is insufficient to induce PD-L1 expression without IL-3 priming. Thus, basophil priming is indispensable for IFN-γ-induced PD-L1 expression in basophils.

## 2. Materials and Methods

### 2.1. Antibodies and Reagents

For flow cytometry, the following antibodies were used. BD Biosciences (Le Pont de Claix, France): HLA-DR-PerCP-Cy™5.5 (Clone: G46-6; 1:50 dilution), CD13-APC (Clone: WM15; 1:50 dilution), CD123-BV421 (Clone: 9F5; 1:50 dilution), CD69-APC/Cy7 (Clone: FN50; 1:50 dilution), CD274 (PD-L1)-FITC/APC/BV421 (Clone: MIH1; 1:50 dilution), CD273 (PD-L2)-PE (Clone: M1H18; 1:50 dilution), CD275 (ICOSL)-PE (Clone: 2D3/B7-H2; 1:50 dilution); Miltenyi Biotec (Paris, France): FcεRIα-PE (Clone: CRA-1; 1:50 dilution); Biolegend (Amsterdam, The Netherlands): CD107a-BV421 (clone H4A3; 1:50 dilution), CD252 (OX40L)-PE (Clone: 11C3.1; 1:50 dilution).

ImmunoTools (Friesoythe, Germany): recombinant human IL-3 (Catalogue: 11,340,037) and recombinant IFN-γ (Catalogue: 11,343,534, lot specific activity: 2 × 10^7^ IU/mg); R&D systems: recombinant thymic stromal lymphopoietin (TSLP) (Catalogue: 1398-TS, lot: IDK0721051); Miltenyi Biotec: IFN-γ (Catalogue: 130-096-484, lot specific activity: ≥2 × 10^7^ IU/mg, recombinant human granulocyte-macrophage colony-stimulating factor (GM-CSF) (Catalogue: 130-093-867, lot specific activity: 1.6 × 10^7^ IU/mg); Sigma-Aldrich (St. Quentin Fallavier, France): Anti-IgE antibodies (Catalogue: I6284, lot: SLCF0672), Lipopolysaccharide (LPS: *E. coli* 055:B5); eBioscience (Paris, France): Fixable viability dye eFluor 506 (1:500 dilution); Lonza (Verviers, Belgium): Serum-free X-VIVO 15 medium.

### 2.2. Purification of Basophils

Ficoll density gradient centrifugation was used to separate peripheral blood mononuclear cells (PBMCs) from the buffy coats of healthy donors (Centre Trinité, L’Établissement Français du Sang, Paris; EFS-INSERM ethical committee permission 18/EFS/033). The negative selection method was used to obtain the purified basophils from the PBMCs by using the Basophil Isolation Kit (Basophil Cell Isolation Kit II, human (Catalogue: 130-092-662) Miltenyi Biotec, Paris, France) according to the manufacturer’s protocol.

### 2.3. Basophil Treatment

Freshly isolated basophils were re-suspended in X-VIVO 15 medium and plated at a concentration of 1 × 10^5^ cells per well in 200 µL (in 96-well plate). Basophils were treated with different concentrations of IFN-γ (100 ng/mL and 1000 ng/mL). In addition, IL-3 at 20 ng/mL concentration was used for priming conditions. Cells were further incubated for 24 h at 37 °C 5% CO_2_. After 24 h of incubation, supernatants were collected and stored at −80 °C for subsequent cytokine quantification. Subsequently, the cells were subjected to surface staining for various markers, such as FcεR1, CD107a, CD13, CD69, and PD-L1, and acquired using LSR II (BD Biosciences). The data were analyzed by using BD FACS DIVA (Version 8.0.1) and FlowJo software (Version 10.4). The gating strategy is presented in Appendix A.

### 2.4. PBMCs Treatment

PBMCs containing basophils were separated from whole blood by the Ficoll density gradient method. PBMCs were re-suspended in complete medium (RPMI 1640 containing 10% fetal bovine serum and 5 units/mL penicillin, and 5 μg/mL streptomycin) and plated at a concentration of 1 × 10^6^ cells in 1000 µL per well (in 24-well plate). PBMCs were treated with different concentrations of IFN-γ (1, 10, 100, and 1000 ng/mL) for 24 h at 37 °C 5% CO_2_. After 24 h of culture, the cells were processed for analysis of various markers on the gated basophil population by flow cytometry. The data were analyzed by using BD FACS DIVA (Version 8.0.1) and FlowJo software (Version 10.4). The gating strategy is presented in Appendix A.

### 2.5. ELISA

Cell-free supernatants from both priming and non-priming conditions of basophils were analyzed for the cytokines IL-4, IL-13, IL-6, CCL3 (MIP-1α), and TSLP (ELISA Ready-SET-Go, eBioscience) according to the manufacturer’s protocol.

### 2.6. Analyses of Expression Levels of IFNGR1, IFNGR2 and CD274

We retrieved the microarray files associated with the published reports of expression profiling of human basophils from http://www.basophil.net and http://162.129.217.250/basophilMicroarrays (accessed on 5 February 2022) [15]. Using these files, we analyzed the expression levels of *IFNGR1*, *IFNGR2*, and *CD274* (PD-L1) in control (CTRL, 0–24 h) or IL-3-treated basophils (TRT, 24–48 h) followed by statistical analysis using the R program (www.r-project.org (accessed on 5 February 2022)). Nonparametric data were compared with a Mann–Whitney test and *p* < 0.05 was considered significant.

### 2.7. Statistical Analysis

As highlighted in the figure legends, the experiments were performed by using cells from several independent donors. Data between two groups were analyzed with Student’s *t*-test followed by the Wilcoxon test. Data among the multiple groups were analyzed by one-way analysis of variance (ANOVA) with Dunnett’s multiple comparison test by using Prism 8 (GraphPad Software Inc., San Diego, CA, USA).

## 3. Results

### 3.1. Lack of Induction of PD-L1 in Human Basophils by IFN-γ

Although IFN-γ-induced PD-L1 expression in macrophages and neutrophils has been shown [8,15], the effect of IFN-γ on human basophils is unknown. To evaluate the effect of IFN-γ on PD-L1 expression in basophils, we first performed experiments with human non-activated primary basophils by treating them with different doses of IFN-γ for 24 h (as earlier studies on macrophages or neutrophils have evaluated the effect of IFN-γ at 18 h time point) [16]. Basophil activation was determined by analyzing the expression of activation markers, such as CD13 and CD69, and degranulation was assessed by CD107a (Figure 1A). Of note, IFN-γ did not induce the expression of either activation or degranulation molecules on the basophils (Figure 1B). Moreover, no significant changes in the expression of PD-L1 were observed on the basophils (Figure 1B). In a similar line, IFN-γ failed to induce the production of the cytokines IL-4, IL-13, IL-6, and chemokine CCL3 by basophils (Figure 1C).

### 3.2. Basophil Priming Is Indispensable for IFN-γ-Induced PD-L1 Expression

As stated above, IFN-γ as such did not directly induce PD-L1 expression. Therefore, we hypothesized that the priming of basophils might play a role in IFN-γ-mediated PD-L1 expression. In this set of experiments, basophils were treated with either IL-3 or a combination of IL-3 and IFN-γ (Figure 2A). As expected, basophils primed with IL-3 displayed enhanced expression of markers associated with basophil activation such as CD13 and CD69. Interestingly, while IL-3 did not affect PD-L1 expression on the basophils, IFN-γ significantly increased the expression of PD-L1 on IL-3-primed basophils (Figure 2B). However, this enhancement in PD-L1 was not associated with an increased expression of activation markers (Figure 2B). On the other hand, IFN-γ increased the secretion of cytokines, such as IL-4 and IL-13 in the primed basophils but IL-6 and CCL3 were not altered (Figure 2C). These results are contrary to the previous findings by another group, particularly pertaining to IL-13 production [16]. Of note, the concentration of IL-3 (20 ng/mL) and IFN-γ (1000 ng/mL) used in the present study was slightly different from the previous studies [16].

### 3.3. IFN-γ-Induced PD-L1 Expression in Basophil Is Legible in PBMCs

To further validate the indispensability of IL-3 priming on IFN-γ-induced PD-L1 expression on the basophils, basophil-containing PBMCs were treated with IFN-γ at different concentrations (1, 10, 100, and 1000 ng/mL) for 24 h. LPS at a concentration of 100 ng/mL was used as a positive control. In these experimental conditions, we did not use IL-3, as T cells in the PBMCs could serve as the source of IL-3. In line with our hypothesis, IFN-γ significantly induced the expression of PD-L1 in basophils (Figure 3A,B). However, it was not associated with an enhancement of activation markers (CD13 and CD69) (Figure 3B). Prolonged stimulation of basophils beyond 24 h with IFN-γ did not further alter the expression of PD-L1 (Appendix A). Interestingly, LPS, which is used as a positive control, also marginally induced the expression of PD-L1, although not significantly (Figure 3B). Perhaps, LPS-mediated marginal PD-L1 expression is of bystander effect; TLR4 (toll-like receptor 4) signaling by LPS is known to stimulate dendritic cells [17], and monocytes that would lead to activation of T cells and a brief production of IFN- γ. Furthermore, natural killer cells could also produce IFN-γ upon stimulation with LPS [18]. Although LPS does not have any direct stimulatory effect on basophils [19], LPS-mediated indirect basophils activation was observed via enhanced CD13 and CD69 expression (Figure 3B) [20]. The effect of IFN-γ is not restricted to one particular source. In fact, IFN-γ from two different commercial sources induced equivalent levels of PD-L1 both in purified basophils (Appendix A) and in untouched basophils (PBMC) (Appendix A). The dose–response study indicated that IFN-γ at 10 ng concentration was sufficient to induce maximum levels of PD-L1 (Appendix A). As IFN-γ-induced higher levels of PD-L1 expression in untouched basophils compared to isolated basophils (Figure 2), the possible influence of other cytokines produced by PBMCs cannot be ruled out.

### 3.4. Basophil Priming by Other Cytokines Is Dispensable for IFN-γ-Induced PD-L1 Expression in Basophils

βc family cytokines such as IL-3, GM-CSF, and IL-5 share a common βc subunit, the signaling subunit of their receptors [21,22]. Although not prominent as IL-3, GM-CSF could induce marginal activation of human basophils [23]. Therefore, we investigated if GM-CSF, like IL-3, could also promote IFN-γ-induced PD-L1 expression in basophils. However, we found that GM-CSF was dispensable for IFN-γ-induced PD-L1 expression (Figure 4), thus suggesting that priming of basophils for IFN-γ-induced PD-L1 expression in human basophils is specific to IL-3.

Next, we investigated the implication of other stimuli, such as anti-IgE and TSLP, on the expression of PD-L1 in basophils. However, both TSLP and anti-IgE failed to induce PD-L1 when combined with either IL-3 or IFN-γ (Figure 5 and data not shown). We confirmed that anti-IgE antibodies were functional as they enhanced the expression of activation (CD69) and degranulation (CD107a) markers on the basophils. Of note, under similar experimental conditions, GM-CSF, anti-IgE, and TSLP did not show any effect on the PD-L2 expression in basophils (Figure 4 and Figure 5 and data not shown for anti-IgE). Possibly, the induction of PD-L2 needs entirely different stimuli.

### 3.5. The Impact of IFN-γ on the Expression of Other Checkpoint Molecules of Basophils

Further, we have explored the effect of IFN-γ on the expression of other checkpoint molecules such as OX40L/CD252 and ICOSL/CD275), and a co-stimulatory molecule CD40) by using untouched basophils. Basophils expressed variable levels of OX40L and ICOSL [24] but IFN-γ stimulation had no impact on their expression (Appendix A). On the other hand, CD40 was not detected or its expression was low in IFN-γ-stimulated basophils.

### 3.6. IL-3 Induces Interferon-Gamma Receptor (IFNGR)2 in Basophils

Based on the above experiments, it is legible that IL-3 priming is indispensable for IFN-γ to induce PD-L1 on the basophils suggesting that IL-3 might affect the expression of IFNGR1/2 and PD-L1. Therefore, expression levels of *IFNGR1*, *IFNGR2*, and *CD274* were analyzed and evaluated from the data set (GSE64664; http://www.basophil.net and http://162.129.217.250/basophilMicroarrays (accessed on 5 February 2022)) [15]. In this data set, all the experiments on basophils had only IL-3 as the intervention. We observed that *IFNGR2* and *CD274* were upregulated whereas *IFNGR1* was downregulated in IL-3-treated basophils (Figure 6 and Figure 7). These results thus provide a mechanistic insight into the role of IL-3 priming in IFN-γ-induced PD-L1 expression on human basophils.

## 4. Discussion

As an immune checkpoint molecule, PD-L1 has gained much attention in several diseases, both for diagnostic and therapeutic purposes. PD-L1 interacts with the programmed death receptor 1 (PD-1) on T cells and induces tolerance or/and exhaustion [25]. More recently, due to the COVID-19 pandemic, research focus on immune checkpoint molecules has increased [26]. As evident from the literature, PD-L1 dysregulation is observed in COVID-19 patients [26]. Moreover, studies have reported the link between PD-L1-expressing basophils and the severity of COVID-19. Remarkably, in severe COVID-19 patients, PD-L1 expression is reduced, and it is well correlated with basophils count [6].

In most situations, IFN-γ performs multiple functions, including recruitment of immune cells, and stimulation of their growth, maturation, and differentiation [27]. It has been reported that macrophages, neutrophils, and various cancer cells express PD-L1 upon IFN-γ stimulation [12,13,15]. Based on these observations, we treated basophils with IFN-γ and our data show that basophils, either unstimulated or stimulated with increasing concentrations of IFN-γ, do not display the expression of PD-L1. Interestingly, priming of basophils with IL-3 led to significantly enhanced expression of PD-L1 on the basophils. In addition, we could reproduce the results in untouched basophils by using basophil-containing PBMCs. Furthermore, IFN-γ-induced PD-L1 expression on basophils is an early event as prolonged stimulation for 48 h did not alter the expression of PD-L1 as compared to 24 h stimulation. It is worth noting that PD-L2 expression is not detected in a comparable experimental setting. Perhaps, an alternative activation of Th2 cells or other factors are required to induce PD-L2 [28]. Despite the fact that IFN-γ has been shown to induce activation of dendritic cells (DCs), macrophages, and monocytes, it induced minimal activation of human basophils. Though the expression of markers associated with basophil activation was enhanced under IL-3 priming conditions, the effect was mainly due to IL-3 rather than IFN-γ. Similarly, to corroborate the effect of IL-3 on IFN-γ induced PD-L1 expression in basophils, we have also explored other priming factors, such as GM-CSF and TSLP, for PD-L1 expression. However, none of these cytokines promoted either PD-L1 or PD-L2 in IFN-γ-stimulated basophils.

OX40L, a TNF superfamily member, is a ligand for OX40. Several studies have reported the importance of OX40-OX40L interaction in shaping the adaptive immune responses, particularly in allergic asthma models [29,30,31,32]. OX40L is expressed on mature dendritic cells, macrophages, and can be induced on B cells, natural killer cells, monocytes, and basophils. Interestingly, OX40L is also expressed by non-immune cells, such as smooth muscle cells [32,33,34,35,36,37]. Moreover, the role of OX40L-expressing basophils in promoting Th2-mediated immune responses is well documented in mouse models of asthma [38]. Whether OX40L on basophils plays a similar role in human pathologies is not known.

Our previous studies have shown that basophils are activated by regulatory T cells in an IL-3 and signal transducer and activator of transcription (STAT)5-dependent manner [24]. Similarly, STAT homodimerization could be responsible for the IFN-γ responses. Consistent with our presumption, IFN-γ enhanced the PD-L1 expression in primed basophils in a dose-dependent manner. The highest expression of PD-L1 was observed for the dose ranging from 100 ng/mL and 1000 ng/mL, which was chosen for further studies. As degranulation markers were not induced in any of the IFN-γ-stimulating conditions, we expect that leukotriene C4 and histamine levels are not altered in these experimental conditions.

Constitutive expression of IFNAR1/2 receptors on basophils has been reported [15,16]. As IL-3 priming plays an important role in IFN-γ-induced PD-L1 on basophils, it is suggested that IL-3 might affect the expression of receptors for IFN-γ on basophils and hence make them receptive for stimulation. In fact, analyses of transcripts for *IFNGR1* and *IFNGR2* by using the data set from the existing source [15] indicated that IL-3 enhances not only *IFNGR2* but also *CD274*. However, this data set has concentrated only on the impact of IL-3. However, in our experiments, we have used IL-3 in combination with IFN-γ. In order to visualize the interplay and expression levels of IFNGR1, IFNGR2, and PD-L1 under the stimulation of both IL-3 and IFN-γ, a new data set is needed.

It is known that basophils mediate Th2 responses in an IL-4-dependent manner, and it is quite evident from the in vitro studies that murine basophils secrete IL-4 and IL-6 that inhibit the IFN-γ, IL-2, and TNF-α expression in T cells [39]. In addition, regulatory T cells secrete IL-3 to activate basophils via the STAT5 signaling cascade, which supports Th2 responses [40]. Basophil functions are regulated by various signals, some requiring priming and others independent of priming. Among various cytokine stimuli, IL-3 is considered to be the most potent stimulator of basophils. In addition, IL-3 priming as reported here and also elsewhere [17,41], enhances the response to other stimuli. IFN-γ acts both in an autocrine and paracrine fashion on immune cells [17,41]. IFN-γ-induced PD-L1 expression has been reported in several cancer cells [14,42,43], and this mechanism is mostly mediated by activating IFNGR1/2-Janus kinase (JAK)/STAT1 pathways [43,44,45,46]. Transcription factors responsible for the IFN-γ-mediated PD-L1 expression and its cross-talk with other signaling pathways are subjects of future investigations [47].

Analyses of PBMCs from severe COVID-19 patients have revealed a dysregulated expression of PD-L1 in various immune cells [26]. However, contradictory results have been published regarding the changes in basophil PD-L1 in COVID-19 patients [6,7]. While one study has reported higher expression of PD-L1 in severe COVID-19 patients compared to milder cases [7], another study has documented reduced PD-L1 in severe and moderate patients as compared to healthy subjects [6]. Of note, in our recent study, we could not detect the expression of PD-L1 on non-activated basophils or basophils primed with IL-3 and—infected or not with SARS-CoV-2 [48]. With the intention that interaction with SARS-CoV-2-infected epithelial cells would affect PD-L1 expression in basophils, we co-cultured basophils with infected human colon epithelial cells Caco-2. Nevertheless, we could not detect any significant changes in the PD-L1 expression in any of the above conditions.

IL-3 acts as a hematopoietic growth factor. A recent study has confirmed the protective role of IL-3 and its potential use as a predictive marker in severe SARS-CoV-2 infections [49]. Of note, our data show that IL-3 is indispensable for IFN-γ-induced PD-L1 expression in basophils. Severe COVID-19 patients display low IL-3 in the circulation [49] that could be responsible for the reduced expression of PD-L1 in basophils reported in one study [6]. In addition, the reduced IL-3 would also affect the survival of basophils and their ability to support protective IgG responses [50]. Though IFN-γ is linked with COVID-19 severity [50], recent data also suggest that enhanced IFN-γ is associated with the recovery-like stage of these patients with a decreased risk of lung fibrosis [51,52]. Importantly, recovered subjects showed an increased expression of PD-L1 in basophils compared to the COVID-19 patients [6]. Based on our data, we suggest that a combination of IL-3 and IFN-γ could be responsible for the enhanced expression of PD-L1 in basophils of recovered COVID-19 patients (Figure 7). Whether PD-L1 expression in basophils plays a role in regulating the T cell inflammatory responses is not known and is the subject of future investigation.

### Perspectives and Outstanding Questions


Does IFN-γ also induce the production or shedding of a soluble form of PD-L1?What is the role of other inflammatory mediators in the induction of PD-L1 on human basophils?Does inflammation-induced PD-L1 on basophils have a role in the pathogenesis of allergy, autoimmunity, cancer, infectious diseases, and others?Does PD-L1 on basophils mediate T cell tolerance similar to DCs? Unlike DCs, basophils do not function as antigen-presenting cells; hence, what are the consequences of T cell responses under similar conditions?


## 5. Conclusions

Based on the above results, it is clear that the response of basophils depends on the type of stimuli they receive. For example, IFN-γ-stimulated basophils express PD-L1 only under IL-3-priming conditions but not in unprimed cells.

## Figures and Tables

**Figure 1 cells-11-00801-f001:**
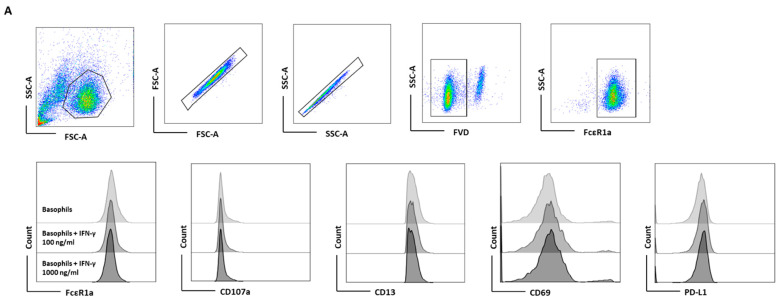
The effect of IFN-γ on the expression of PD-L1 in primary human basophils. Basophils (0.1 × 10^6^ cells/200 μL/96-well plate) isolated from PBMCs of healthy donors were cultured with or without IFN-γ at 100 or 1000 ng/mL. Basophil phenotype was evaluated by flow cytometry after 24 h. (**A**) Dead cells were excluded with the help of fixable viability dye (FVD) and live basophils were gated. Representative histogram overlays displaying the expression pattern of FcεRI, CD107a, CD13, CD69, and PD-L1. (**B**) Expression of FcεRI, CD107a, CD13, CD69, and PD-L1 on the basophils (% positive cells and median fluorescence intensities (MFI), mean ± SD; *n* = 5 independent donors with three independent experiments). (**C**) The amount (pg/mL) of secreted IL-4, IL-13, CCL3 and IL-6 in the cell-free supernatant from the above experiments (mean ± SD, *n* = 5 to 8 independent donors from three independent experiments). ns, not significant, ** *p* < 0.01, one-way ANOVA Friedman test with Dunn’s multiple comparisons post-test.

**Figure 2 cells-11-00801-f002:**
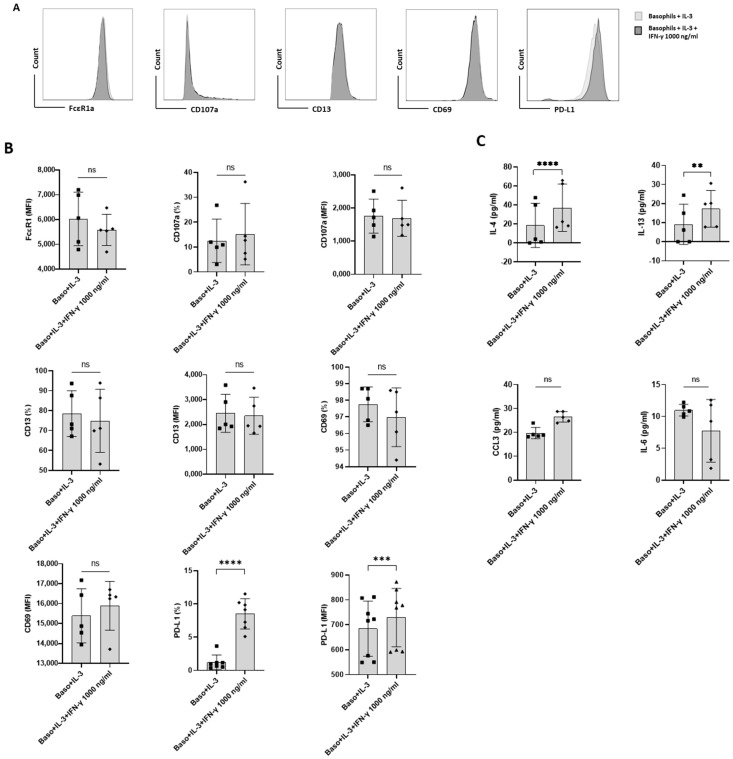
The effect of IFN-γ on PD-L1 expression in primed primary human basophils. Basophils (0.1 × 10^6^ cells/200 μL/96-well plate) isolated from PBMCs of healthy donors were cultured with or without IFN-γ at 1000 ng/mL. Simultaneously, basophils were primed with IL-3 along with IFN-γ treatment. Basophil phenotype was evaluated by flow cytometry after 24 h. (**A**) Gating strategy and representative histogram overlays displaying the expression pattern of FcεR1, CD107a, CD13, CD69, and PD-L1. (**B**) Expression of FcεR1, CD107a, CD13, CD69, and PD-L1 on basophils (% positive cells and median fluorescence intensities (MFI), mean ± SD; *n* = 5–8 independent donors with three independent experiments). (**C**) The amount (pg/mL) of secreted IL-4, IL-13, CCL3 and IL-6 in the cell-free supernatant from the above experiments. The data were presented as the mean ± SD and were from 5–8 independent donors and three independent experiments). ns, not significant, ** *p* < 0.01, *** *p* < 0.001, **** *p* < 0.0001, paired Wilcoxon test.

**Figure 3 cells-11-00801-f003:**
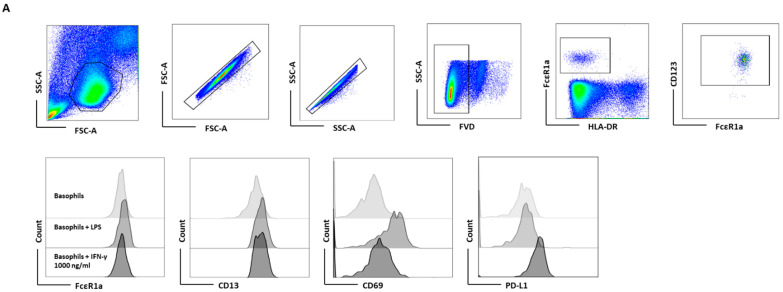
IFN-γ-induced PD-L1 expression in basophil is legible in PBMCs. Basophil-containing PBMCs (1 × 10^6^ cells/mL/24-well plate) from the healthy donors were cultured with or without IFN-γ at 1000 ng/mL or LPS at 100 ng/mL for 24 h. After incubation, cells’ phenotype was evaluated by flow cytometry. (**A**) Gating strategy and representative histogram overlays are displaying the expression pattern of FcεR1, CD107a, CD13, CD69, and PD-L1 on the basophils. (**B**) Expression of FcεR1, CD13, CD69, and PD-L1 on the basophils (% positive cells and median fluorescence intensities (MFI), mean ± SD; *n* = 4–11 independent donors from three independent experiments). ns, not significant, * *p* < 0.05, ** *p* < 0.01, *** *p* < 0.001, one-way ANOVA Friedman test with Dunn’s multiple comparisons post-test.

**Figure 4 cells-11-00801-f004:**
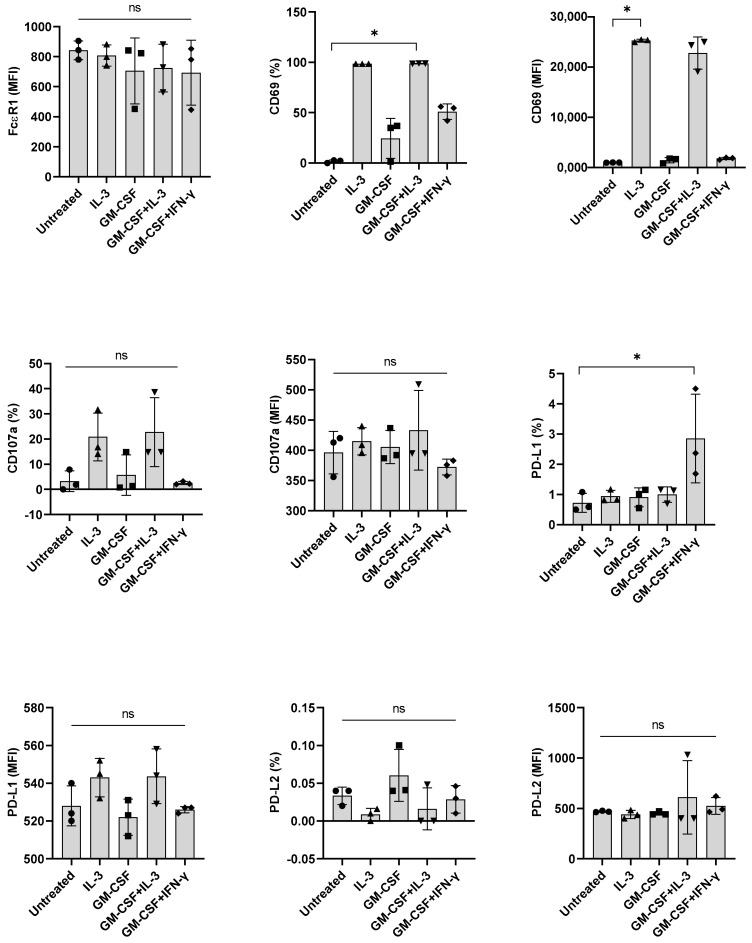
The effect of GM-CSF on the expression of PD-L1, induced by IFN-γ in primary human basophils. Basophils (0.1 × 10^6^ cells/200 μL/96-well plate) isolated from PBMCs of healthy donors were cultured with either IL-3, GM-CSF (5 ng/mL), GM-CSF + IL-3 or GM-CSF + IFN-γ. Basophil phenotype was evaluated by flow cytometry after 24 h. Expression of FcεRI, CD69, CD107a, PD-L1, and PD-L2 on the basophils (% positive cells and median fluorescence intensities (MFI), mean ± SD; *n* = 3 independent donors from three independent experiments) was presented. ns, not significant, * *p* < 0.05, one-way ANOVA Friedman test with Dunn’s multiple comparisons post-test.

**Figure 5 cells-11-00801-f005:**
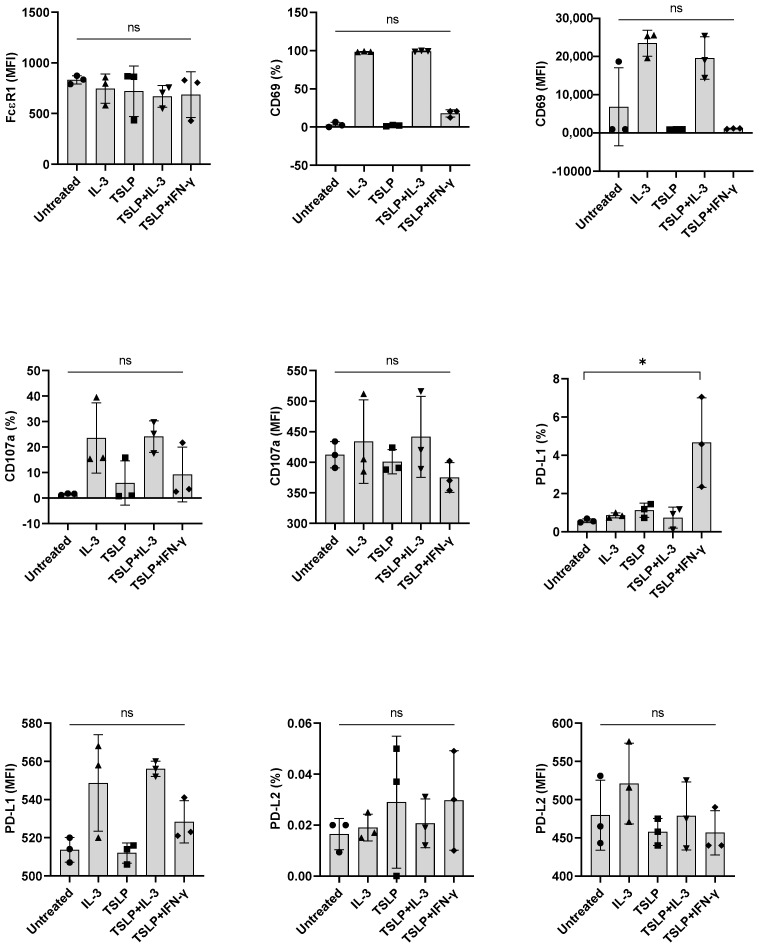
The effect of TSLP on the expression of PD-L1 induced by IFN-γ in primary human basophils. Basophils (0.1 × 10^6^ cells/200 μL/96-well plate) isolated from PBMCs of healthy donors were cultured with either IL-3, TSLP (10 ng/mL), TSLP + IL-3 or TSLP + IFN-γ. Basophil phenotype was evaluated by flow cytometry after 24 h. Expression of FcεRI, CD69, CD107a, PD-L1, and PD-L2 on the basophils (% positive cells and median fluorescence intensities (MFI), mean ± SD; *n* = 3 independent donors from three independent experiments) was presented. ns, not significant, * *p* < 0.05, one-way ANOVA Friedman test with Dunn’s multiple comparisons post-test.

**Figure 6 cells-11-00801-f006:**
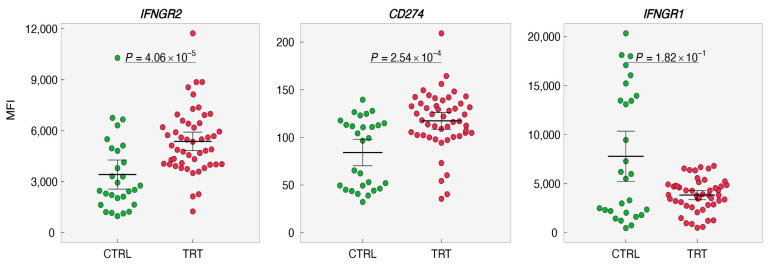
Impact of IL-3 priming on the expression of IFNGRs and PD-L1 on human basophils. The expression levels of *IFNGR1*, *IFNGR2*, and *CD274* (PD-L1) in control (CTRL, 0–24 h) or treated basophils (TRT,24–48 h) were retrieved from (http://www.basophil.net and http://162.129.217.250/basophilMicroarrays; accessed on 5 February 2022) [15]. Statistical analyses were carried out using the R program (www.r-project.org (accessed on 5 February 2022)). Measurement data between two groups were performed using a nonparametric Mann–Whitney test. *p* < 0.05 was considered significant. MFI—median fluorescence intensity.

**Figure 7 cells-11-00801-f007:**
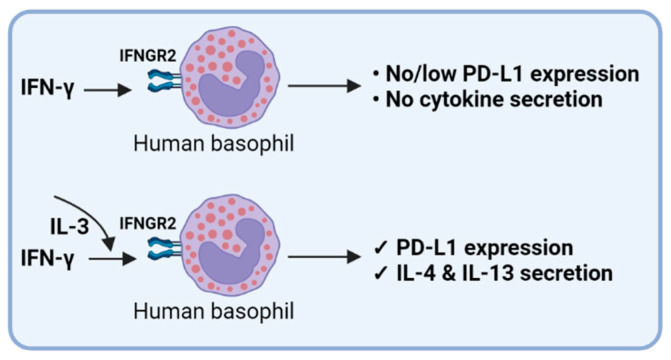
IFN-γ-induced PD-L1 expression in human basophils.

## Data Availability

Data reported in this study are available in the manuscript.

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
