# Peer review of "IFN-γ Induces PD-L1 Expression in Primed Human Basophils"

_cells, 2022, doi:10.3390/cells11050801_

Round 1

Reviewer 1 Report

In this manuscript by Bonam, Chauvin and Bayry, the authors attempt to elucidate some of the mechanisms behind PD-L1 expression by basophils. I am charmed by how this brief report tries to address cells that are normally consequently marginalized, and how authors try to elucidate whether basoph8ils express PD-L1 without going into the functional consequences. I do not dislike that authors leave the story “unfinished”, but I have some concerns regarding potential reproducibility or representativeness of the data. How many independent experiments were performed for the data displayed in each figure? The maximum amount of independent datapoints is 3 to 5 in all figures/graphs, to me this sounds like 1 or 2 experiments at most. I would like to ask the authors to repeat the experiments performed for the data in figures 1-3 as to determine whether this is representative, or, if this is representative of, let’s say 3 experiments, state that experiments were performed 3 times with X donors and that the data displayed is representative in the figure legends of corresponding figures. As new experiments might be involved, I think this consitutes a major revision. However, I do think that with these extra experiments, the paper is of high interest for readers of Cells and should be publishable.

In my opinion, the two main points authors should adress are amount of replicates/indipendent experiments, and indicating how gating on markers is performed (see also later in my review report).

General

  • Remove COVID-19 and SARS-CoV-2 as keywords, as this manuscript is, at best, marginally related, and more of a general interest, if authors cannot add the unpublished SARS-CoV-2 data that they discuss in the discussion.

Introduction

Line 18 – start sentence with “Despite”.

Line 20 – remove and before “in the pathogenesis of asthma” OR remove the comma after asthma.

Line 27 – several stimuli can induce PD-L1 expression – I suggest to use the plural here too.

Line 28 – add expression after PD-L1

Line 29 – pathogen-derived

Line 33 – render is out of place, suggest to use “result in”

Materials & Methods

Line 38 – “For flow cytometry,”

Line 38 – great that you add both fluorophore and clone of the antibodies, please also add the used dilutions. Also for the fixable viability dye.

Line 52 – according to manufacturer’s protocol? If yes, please add, if deviated, please elaborate. Please also add the type of isolation kit.

Line 56 – please provide the IU/mL for the specific lot of IFN-g used if available for reproducibility in other labs. Same for IL-3.

Line 61 – Please add version numbers for DIVA and Flow-Jo (same for line 70-71).

Line 64 – remove the before whole blood.

Line 67 – how many uL/well (i.e. how many cells/well)

Line 69 – remove the before analyses, and use analysis instead.

Line 74 – according to manufacturer’s protocol? If yes, please add, if deviated, please elaborate.

Results

Line 83 – is studies = has been shown

Line 85 – towards  = on

Line 85 – first series of experiments were performed = we first performed experiments

Line 86 – naïve? Do you mean untouched, as a by-product of the MACS isolation being a negative selection? Or do you simply mean rested/non-activated? Please say so. In general, naïve is only used for antigen-dependent responses of B and T cells.

Line 92 – failed to induce the production of the cytokines IL-4 and IL-13 by basophils.

Figure 1 – did authors include a positive control for stimulation, i.e. IL-3 /IL-33 / IgE or any other basophil activator? Otherwise, one could argue that the staining might not have been optimal. If yes, please add this data, maybe as a supplementary figure.

A – please add where you put the gate on what you consider positive cells.

Legend: please add that you exclude dead cells via a fixable viability dye (FVD). As indicated above, please add representativeness of data (n=5 from X experiments).

Line 105 – affect = induce

Line 106 – presumed = hypothesized

Line 109 – like = such as, and please put in the same order as in the figure (“such as CD13 and CD69”)

Line 115 – nevertheless = of note

Figure 2

A – please add where you put the gate on what you consider positive cells, as in B I cannot decipher how this minimal shift in PD-L1 is considered “significant”, nor how it results in an increase in 20 percent. Perhaps this is not the most representative donor for PD-L1 expression?

Legend: As indicated above, please add representativeness of data (n=X from X experiments).

Line 129 – “To further validate”, towards = on

Line 130 – please add expression after PD-L1

Line 131 – add a full stop after 24 h, start a new sentence with “LPS at a concentration of 100 ng/mL was used as…”

Line 132 – would = could

Line 139 – monocytes/DCs would not be able to induce T cell activation in the absence of antigens, perhaps only marginally, and that would only result in a brief production of IFN-g by T cells (see e.g. https://doi.org/10.1111/sji.13019 on how bystander T cell responses are only resulting in marginal cytokine production rather than full-blown activation). I would rather think this is due to IFN-g production by NK cells after LPS sensing (see https://dx.doi.org/10.3389%2Ffimmu.2013.00011) – please amend this section!

Line 141 – remove the.

Figure 3 –

A – please add where you put the gate on what you consider positive cells

Legend: As indicated above, please add representativeness of data (n=X from X experiments).

Supplementary figures – some of these figures have very few datapoints. For reproducibility and credibility, authors should include data in all figures (including main figures) of at least 3 donors from at least 2 independent experiments. Otherwise, the data cannot be considered representative.

Discussion

Line 155 – “viable”? please remove.

Line 157 – remove the

Line 166 – why not add this data to this paper? Or add it as a supplementary figure? If authors are not able to, please remove the part describing unpublished work, or supply the data to the journal editor to validate the findings.

Line 173 – “and differentiation”

Line 175 – remove the

Line 178 – leads, or led

Line 184 – should be = is

Line 196 – Consistent

Line 205 – this is something I have been wondering when reading this paper. Several datasets are out there describing profiling of basophils, such as https://dx.doi.org/10.1371%2Fjournal.pone.0126435 ; this particular paper even describes differences between untreated and IL-3-treated basophils. Alternatively, authors could use tools such as the ImmGen’s Gene Skyline, which shows that the average expression is 800 normalized units for IFNGR1, while CD274 (PD-L1) is expressed at 1000 units in blood-derived basophils. Authors should analyze these datasets and determine whether IFNGR1/2 are upregulated in basophils. This could even be added as a supplementary figure.

Line 209 – IL-2 and TNF-a

Line 210 – secrete IL-3 to activate basophils via a STAT5 signaling cascade, which supports Th2 responses

Line 212 – stimulators = stimuli

Line 213 - considered to be the

Line 214 – please add reference(s) to other papers after elsewhere

Line 214 – the = an autocrine

Line 215 – remove the before immune cells.

Line 221 – or does it also induce the production or shedding of a soluble form of PD-L1?

Line 227 – Does PD-L1 expressed by basophils mediate T cell tolerance, similar to DCs?
Of note, basophils express class I (as do all other nucleated cells) and thus are antigen-presenting cells, just not in the classical sense that they can serve as primary stimulators of immune responses due to lacking the right costimulatory receptors. However, it has been shown that they can actually serve as APCs, see https://dx.doi.org/10.1016%2Fj.coi.2010.01.012 for an excellent review on basophils and Th2 responses.

Author Response

In this manuscript by Bonam, Chauvin and Bayry, the authors attempt to elucidate some of the mechanisms behind PD-L1 expression by basophils. I am charmed by how this brief report tries to address cells that are normally consequently marginalized, and how authors try to elucidate whether basoph8ils express PD-L1 without going into the functional consequences. I do not dislike that authors leave the story “unfinished”, but I have some concerns regarding potential reproducibility or representativeness of the data. How many independent experiments were performed for the data displayed in each figure? The maximum amount of independent datapoints is 3 to 5 in all figures/graphs, to me this sounds like 1 or 2 experiments at most. I would like to ask the authors to repeat the experiments performed for the data in figures 1-3 as to determine whether this is representative, or, if this is representative of, let’s say 3 experiments, state that experiments were performed 3 times with X donors and that the data displayed is representative in the figure legends of corresponding figures. As new experiments might be involved, I think this consitutes a major revision. However, I do think that with these extra experiments, the paper is of high interest for readers of Cells and should be publishable.

In my opinion, the two main points authors should adress are amount of replicates/indipendent experiments, and indicating how gating on markers is performed (see also later in my review report).

We understand the reviewer viewpoint. Therefore, the experiments were repeated wherever appropriate. As experiments are conducted by using the primary immune cells from the healthy donors, each experiment was repeated with indpendent donors in independent experiments. Also, we used independent donors with variable genetic and environmental background (unlike experimental animals where inbred, single strain mice are commonly used).  The details are included in the respective figure legends.

The gating strategy is presented as supplementary Figure S1. 

General

  • Remove COVID-19 and SARS-CoV-2 as keywords, as this manuscript is, at best, marginally related, and more of a general interest, if authors cannot add the unpublished SARS-CoV-2 data that they discuss in the discussion.

In line with the suggestion of the reviewer, we have removed COVID-19 and SARS-CoV-2 as keywords

Introduction

Line 18 – start sentence with “Despite”.

Inserted (Line 26)

Line 20 – remove and before “in the pathogenesis of asthma” OR remove the comma after asthma.

Corrected (comma has been removed) (Line 28)

Line 27 – several stimuli can induce PD-L1 expression – I suggest to use the plural here too.

Corrected (Line 35)

Line 28 – add expression after PD-L1

Corrected (Line 36)

Line 29 – pathogen-derived

Corrected (Line 37)

Line 33 – render is out of place, suggest to use “result in”

Corrected appropriately (Line 42)

Materials & Methods

Line 38 – “For flow cytometry,”

Corrected (Line 57)

Line 38 – great that you add both fluorophore and clone of the antibodies, please also add the used dilutions. Also for the fixable viability dye.

Thank you, the revised manuscript contains the dilution of each antibody and fixable viability dye. (Line 57-64)

Line 52 – according to manufacturer’s protocol? If yes, please add, if deviated, please elaborate. Please also add the type of isolation kit.

The necessary details have been added to the revised manuscript (Lines 80-81)

Line 56 – please provide the IU/mL for the specific lot of IFN-g used if available for reproducibility in other labs. Same for IL-3.

Lot specific activity in IU/mL is represented for IFN-g and other cytokines. For IL-3, lot specific activity in IU/mL is not available rather ED50 information is provided by the supplier (<0.1ng/ml) (Lines 65-70)

Line 61 – Please add version numbers for DIVA and Flow-Jo (same for line 70-71).

We have added version numbers for DIVA and Flow-Jo (Lines 92, 103)

Line 64 – remove the before whole blood.

Removed  (Line 96)

Line 67 – how many uL/well (i.e. how many cells/well)

Required information provided (Line 99)

Line 69 – remove the before analyses, and use analysis instead.

Corrected (Line 101)

Line 74 – according to manufacturer’s protocol? If yes, please add, if deviated, please elaborate.

 Added (Line 114)

Results

Line 83 – is studies = has been shown

Corrected (Lines 135-136)

Line 85 – towards  = on

Corrected (Line 137)

Line 85 – first series of experiments were performed = we first performed experiments

Corrected (Line 137)

Line 86 – naïve? Do you mean untouched, as a by-product of the MACS isolation being a negative selection? Or do you simply mean rested/non-activated? Please say so. In general, naïve is only used for antigen-dependent responses of B and T cells.

Replaced the “naïve” with “non-activated” (Line 138)

Line 92 – failed to induce the production of the cytokines IL-4 and IL-13 by basophils.

Modified (Line 145)

Figure 1 – did authors include a positive control for stimulation, i.e. IL-3 /IL-33 / IgE or any other basophil activator? Otherwise, one could argue that the staining might not have been optimal. If yes, please add this data, maybe as a supplementary figure.

We have conducted the study with many basophil priming factors or stimuli, such as IL-3, anti-IgE, GM-CSF, and TSLP. Data are presented in the article under Figures 2, 4 and 5

A – please add where you put the gate on what you consider positive cells.

We assume that our studies are very familiar to biologists. However, for understanding, the gating strategy has been attached as a supplementary Figure S1.

Legend: please add that you exclude dead cells via a fixable viability dye (FVD). As indicated above, please add representativeness of data (n=5 from X experiments).

Added

Line 105 – affect = induce

Modified (Line 168)

Line 106 – presumed = hypothesized

Modified (Line 169)

Line 109 – like = such as, and please put in the same order as in the figure (“such as CD13 and CD69”)

Amended (Line 172-173)

Line 115 – nevertheless = of note

Modified (Line 179)

Figure 2

A – please add where you put the gate on what you consider positive cells, as in B I cannot decipher how this minimal shift in PD-L1 is considered “significant”, nor how it results in an increase in 20 percent. Perhaps this is not the most representative donor for PD-L1 expression?

The gating strategy has been attached as a supplementary Figure S1. The revised manuscript shows the “shift in PD-L1 expression” compared to the non-treated.

Legend: As indicated above, please add representativeness of data (n=X from X experiments).

Necessary information is provided in the figure legends.

Line 129 – “To further validate”, towards = on

Modified (Line 209)

Line 130 – please add expression after PD-L1

Added (Lines 209-210)

Line 131 – add a full stop after 24 h, start a new sentence with “LPS at a concentration of 100 ng/mL was used as…”

Modified (Line 211)

Line 132 – would = could

Modified (Line 225)

Line 139 – monocytes/DCs would not be able to induce T cell activation in the absence of antigens, perhaps only marginally, and that would only result in a brief production of IFN-g by T cells (see e.g. https://doi.org/10.1111/sji.13019 on how bystander T cell responses are only resulting in marginal cytokine production rather than full-blown activation). I would rather think this is due to IFN-g production by NK cells after LPS sensing (see https://dx.doi.org/10.3389%2Ffimmu.2013.00011) – please amend this section!

The text was amended (Lines 233-234)

Line 141 – remove the.

Removed (Lne 235)

Figure 3 –

A – please add where you put the gate on what you consider positive cells

The gating strategy has been attached as a supplementary Figure S1.

Legend: As indicated above, please add representativeness of data (n=X from X experiments).

 Necessary information is provided in the figure legend.

Supplementary figures – some of these figures have very few datapoints. For reproducibility and credibility, authors should include data in all figures (including main figures) of at least 3 donors from at least 2 independent experiments. Otherwise, the data cannot be considered representative.

 Essential experiments were repeated and figures were modified accordingly.

Discussion

Line 155 – “viable”? please remove.

Removed (Line 327)

Line 157 – remove the

Removed (Line 329)

Line 166 – why not add this data to this paper? Or add it as a supplementary figure? If authors are not able to, please remove the part describing unpublished work, or supply the data to the journal editor to validate the findings.

The discussed study has been accepted for the publication in “Frontiers in Immunology”. Therefore, we have provided the respective citation. (The article will be live on the 24th February)

Line 173 – “and differentiation”

Amended (Line 337)

Line 175 – remove the

Removed (Line 339)

Line 178 – leads, or led

Amended (Line 342)

Line 184 – should be = is

Amended (Line 347)

Line 196 – Consistent

Corrected (Line 377)

Line 205 – this is something I have been wondering when reading this paper. Several datasets are out there describing profiling of basophils, such as https://dx.doi.org/10.1371%2Fjournal.pone.0126435 ; this particular paper even describes differences between untreated and IL-3-treated basophils. Alternatively, authors could use tools such as the ImmGen’s Gene Skyline, which shows that the average expression is 800 normalized units for IFNGR1, while CD274 (PD-L1) is expressed at 1000 units in blood-derived basophils. Authors should analyze these datasets and determine whether IFNGR1/2 are upregulated in basophils. This could even be added as a supplementary figure.

Thank you very much for your pertinent suggestion. As suggested, the aforementioned datasets were retrieved and analysed for the IFNGR1/2 and PD-L1 expression. Interestingly, we could see the upregulation of IFNGR2 and CD274 transcripts in IL-3-treated basophils. The data are included in Figure 6.

Line 209 – IL-2 and TNF-a

Amended (Line 396)

Line 210 – secrete IL-3 to activate basophils via a STAT5 signaling cascade, which supports Th2 responses

Amended (Line 396-397)

Line 212 – stimulators = stimuli

Corrected (Line 399)

Line 213 - considered to be the

Corrected (Line 400)

Line 214 – please add reference(s) to other papers after elsewhere

Added (Line 401)

Line 214 – the = an autocrine

Corrected (Line 401)

Line 215 – remove the before immune cells.

Removed (Line 402)

Line 221 – or does it also induce the production or shedding of a soluble form of PD-L1?

Amended (Line 449)

Line 227 – Does PD-L1 expressed by basophils mediate T cell tolerance, similar to DCs?
Of note, basophils express class I (as do all other nucleated cells) and thus are antigen-presenting cells, just not in the classical sense that they can serve as primary stimulators of immune responses due to lacking the right costimulatory receptors. However, it has been shown that they can actually serve as APCs, see https://dx.doi.org/10.1016%2Fj.coi.2010.01.012 for an excellent review on basophils and Th2 responses.

We respectfully disagree with the reviewer. Work from other labs (Allergy (2012) 67:601-8; Allergy (2012) 67:593-600) and ours (Sci Rep (2013) 3:1188; Hum Immunol (2015) 76:176-80; Haematologica (2017) 102:e233-e7) have repeatedly shown that both circulating and tissue-specific (like spleen, lymphnodes) human basophils do not act as APC. Basophils lack HLA-DR and co-stimulatory molecules, and are inept at inducing T cell polarization.

Reviewer 2 Report

This is a straightforward in vitro investigation, showing Th1 cytokine IFN-γ can upregulate the cell surface expression of immune checkpoint molecule PD-L1 on IL-3 primed human basophils. Although the results have been clearly presented, there are several comments as below.

(1) Apart from PDL1, whether other immune check point proteins such as CD40, ICOSL, OX40L may also be regulated on primed basophils upon IFN-γ stimulation. This should be further discussed.

(2) Apart from Th2 cytokine IL-4 and IL-13, if other basophil releasing cytokines such as IL-6, CCL3 and TSLP etc. may also be induced from primed basophils upon IFN-γ stimulation?

(3) Since Th2-related TSLP and IL-33 are the well characterized basophil activating cytokines, whether these 2 basophil activating cytokines have similar effect as IFN-γ on primed basophils?

(4) In Figure 2, the concentration of IFN-γ (1000 ng/ml) used seems to be high. Please clarify if this concentration is physiological relevant.

(5) What are the potential intracellular signaling mechanism (e.g. STAT etc.) by which IFN-γ can upregulate PDL-1 expression on IL-3 primed basophils?

(6) As mentioned in the manuscript, there is a link between PD-L1-expressing basophils and severity of COVID-19. It has been shown that low basophil number can account for the decreased protection in COVID-19 patients. Since PDL-1 upregulation can also be induced by SARS-COV-2, what is the immunopathological role of the increased PDL1 in primed basophils upon activation by IFN-γ in COVID-19 patients? It needs further elaboration and discussion.

Author Response

This is a straightforward in vitro investigation, showing Th1 cytokine IFN-γ can upregulate the cell surface expression of immune checkpoint molecule PD-L1 on IL-3 primed human basophils. Although the results have been clearly presented, there are several comments as below.

(1) Apart from PDL1, whether other immune check point proteins such as CD40, ICOSL, OX40L may also be regulated on primed basophils upon IFN-γ stimulation. This should be further discussed.

In view of the reviewer suggestion, the experiments were conducted on untouched basophils to evaluate the expression of CD40, ICOSL, and OX40L. Except for CD40, the expression other immune check point proteins was seen in the basophils but their expresion was not altered in IFN treated conditions. The results are presented under 3.5 and supplement Figure S5.

(2) Apart from Th2 cytokine IL-4 and IL-13, if other basophil releasing cytokines such as IL-6, CCL3 and TSLP etc. may also be induced from primed basophils upon IFN-γ stimulation?

Thank you for the suggestion. We have evaluated the suggested cytokines and chemokines. The revised manuscript contains the updated information (Figure 1C and 2C). Though basophils produced IL-6 and CCL3, IFN-γ stimulation did not alter them. On the other hand, we could not detect TSLP.

(3) Since Th2-related TSLP and IL-33 are the well characterized basophil activating cytokines, whether these 2 basophil activating cytokines have similar effect as IFN-γ on primed basophils?

Considering the reviewer suggestion, we have performed the experiments with TLSP and other basophil activating factors, such as GM-CSF and anti-IgE with IFN-γ or IL-3. Any of these stimuli did not induce PD-L1 expression. Regarding IL-33, we did not recive the cytokine from the commercial supplier within the deadline to perform the experiments. However, in our previous report, we found that IL-33 induces only marginal activation of  basophils (Galeotti et al JACI 2019) Therfore, we do not expect any effect of IL-33 on primed basophils

(4) In Figure 2, the concentration of IFN-γ (1000 ng/ml) used seems to be high. Please clarify if this concentration is physiological relevant.

To our knowledge, the literature describes the presence of 250-1000 ng/ml of IFN-γ in plasma. However, there is a huge variation in detection methods (https://doi.org/10.1149/2.0271605jes, https://doi.org/10.1016/j.bios.2004.11.008). Also, T cells produce a huge amount of IFN-γ when they are activated.

(5) What are the potential intracellular signaling mechanism (e.g. STAT etc.) by which IFN-γ can upregulate PDL-1 expression on IL-3 primed basophils?

Although in cancer cells, the IFN-γ induced JAK2/STAT1 pathways are known to induce PD-L1, the implication of the same pathway in primed basophils is not known. We have included these points in the discussion. (Lines 402-404)

(6) As mentioned in the manuscript, there is a link between PD-L1-expressing basophils and severity of COVID-19. It has been shown that low basophil number can account for the decreased protection in COVID-19 patients. Since PDL-1 upregulation can also be induced by SARS-COV-2, what is the immunopathological role of the increased PDL1 in primed basophils upon activation by IFN-γ in COVID-19 patients? It needs further elaboration and discussion.

Thank you for the valuable suggestion. We have elaborated the above points in the discussion (Lines 408-441).

Round 2

Reviewer 1 Report

I would like to thank the authors for taking my suggestions in consideration and incorporating most of them. Congratulations on this beautiful manuscript. I recommend acceptance to the editors.

Reviewer 2 Report

The manuscript has been revised properly according to reviewer’s comments, suggestions and questions. I recommend the revised manuscript can be accepted to be published in Cells.